# Study on the Enhanced Remediation of Petroleum-Contaminated Soil by Biochar/g-C_3_N_4_ Composites

**DOI:** 10.3390/ijerph19148290

**Published:** 2022-07-07

**Authors:** Hongyang Lin, Yang Yang, Zhenxiao Shang, Qiuhong Li, Xiaoyin Niu, Yanfei Ma, Aiju Liu

**Affiliations:** 1School of Agricultural Engineering and Food Science, Shandong University of Technology, Zibo 255049, China; linhongyang2021@163.com (H.L.); 21403010279@stumail.sdut.edu.cn (Y.Y.); 2School of Resources and Environmental Engineering, Shandong University of Technology, Zibo 255049, China; alphazx@sdut.edu.cn (Z.S.); zbnxy@sdut.edu.cn (X.N.); 3School of Materials Science and Engineering, Shandong University of Technology, Zibo 255049, China; qhli@sdut.edu.cn

**Keywords:** biochar, adsorption, photocatalysis, synergy, TPH, soil

## Abstract

This work developed an environmentally-friendly soil remediation method based on BC and g-C_3_N_4_, and demonstrated the technical feasibility of remediating petroleum-contaminated soil with biochar/graphite carbon nitride (BC/g-C_3_N_4_). The synthesis of BC/g-C_3_N_4_ composites was used for the removal of TPH in soil via adsorption and photocatalysis. BC, g-C_3_N_4_, and BC/g-C_3_N_4_ have been characterized by scanning electron microscopy (SEM), Brunauer–Emmett–Teller surface area analyzer (BET), FT-IR, and X-ray diffraction (XRD). BC/g-C_3_N_4_ facilitates the degradation due to reducing recombination and better electron-hole pair separation. BC, g-C_3_N_4_, and BC/g-C_3_N_4_ were tested for their adsorption and photocatalytic degradation capacities. Excellent and promising results are brought out by an apparent synergism between adsorption and photocatalysis. The optimum doping ratio of 1:3 between BC and g-C_3_N_4_ was determined by single-factor experiments. The removal rate of total petroleum hydrocarbons (TPH) by BC/g-C_3_N_4_ reached 54.5% by adding BC/g-C_3_N_4_ at a dosing rate of 0.08 g/g in a neutral soil with 10% moisture content, which was 2.12 and 1.95 times of BC and g-C_3_N_4_, respectively. The removal process of TPH by BC/g-C_3_N_4_ conformed to the pseudo-second-order kinetic model. In addition, the removal rates of different petroleum components in soil were analyzed in terms of gas chromatography–mass spectrometry (GC-MS), and the removal rates of nC_13_-nC_35_ were above 90% with the contaminated soil treated by BC/g-C_3_N_4_. The radical scavenger experiments indicated that superoxide radical played the major role in the photocatalytic degradation of TPH. This work definitely demonstrates that the BC/g-C_3_N_4_ composites have great potential for application in the remediation of organic pollutant contaminated soil.

## 1. Introduction

In recent years, with the development of society, China’s demand for petroleum and other products is increasing [1], but the problems of leakage in the process of petroleum exploitation, transportation, refining, processing, and use are becoming more and more serious [2,3]. A large amount of petroleum accumulates in soil, which inhibits the activity of indigenous microorganisms in soil, and destroys the ecological balance of the soil [4,5]. The petroleum hydrocarbons in soil will seep into the groundwater by migration and cause the pollution of groundwater, which poses a threat to human health [6,7].

At present, the remediation methods of petroleum-contaminated soil mainly include adsorption [8], incineration [9], chemical oxidation [10], leaching [11], phytoremediation [12], and the microbial remediation method [13,14,15]. Among them, the adsorption method is widely preferred because of its simple operation, low price, and relatively good adsorption effect [16,17]. BC is a pyrogenic carbonaceous material produced through the pyrolysis (350–700 °C) of agriculture and forestry biomass, food processing waste, animal manure, and other industrial waste biomass under an oxygen-free environment [18,19,20], which has a wide range of sources and a cheaper cost [21]. It has good adsorption and special photoelectric properties [22]. However, BC has some disadvantages in practical application, such as poor selective adsorption, easy saturation, easy desorption, insufficient long-acting performance, and so on [23]. Thus, some researchers focused on enhancing the removal capacity of BC by modifying its surface properties. For instance, Gurav et al. [20] modified pinewood BC with coconut oil to improve the hydrophobicity of BC, thereby increasing the adsorption capacity of BC on crude oil. The maximum adsorption capacity of the modified BC to crude oil reached 5315 mg/g in 60 min. Nguyen et al. [24] modified BC with FeCl_3_, AlCl_3_, and CaCl_2_ to increase the number of surface functional groups, and the maximum chromium removal rate was 96.8%. Liu et al. [25] pretreated camellia oleifera shell with aminosulfonic acid to obtain the modified BC with a larger specific surface area and more functional groups, and the maximum adsorption capacity of tetracycline reached 412.95 mg/g.

In recent years, the photocatalytic degradation technology based on visible light irradiation has been diffusely recognized as a highly-efficient, low-cost, and environmentally-friendly method. However, the majority of traditional photocatalysts are metal-based materials, which can cause secondary pollution of the soil when they enter it [26]. g-C_3_N_4_ is a new type of non-metallic semiconductor photocatalytic material [27]. It has a large specific surface area, suitable energy band, and high photosensitivity, and can be regarded as a promising photocatalyst in the remediation of soil [28,29,30]. For example, Luo et al. [31] prepared g-C_3_N_4_ by three different precursors, including urea, dicyandiamide, and melamine, via the conventional thermal polymerization method. Under visible light or natural sunlight, g-C_3_N_4_ synthesized by three different precursors can remarkably reduce the toxicity of phenanthrene-contaminated soil. However, g-C_3_N_4_ also has the defect that the photogenerated charge carriers are recombined rapidly, which, consequently, seriously affects the application of g-C_3_N_4_ in the field of photocatalysis [32]. The BC/g-C_3_N_4_ composites were prepared by loading g-C_3_N_4_ onto the surface of BC, which could greatly improve the photocatalytic ability of g-C_3_N_4_. BC/g-C_3_N_4_ has the potential application in the remediation of soil because of its enhanced photocatalytic activities and lower toxicity. The enhanced photocatalytic activities of BC/g-C_3_N_4_ should be attributed to two aspects. On the one hand, BC could be used as effective electron transfer channels and acceptors to improve the separation of photogenerated electron–hole pairs. On the other hand, the large surface area of biochar can enrich pollutants while providing sufficient catalytic sites for photocatalytic degradation [33]. Li and Lin [34] successfully prepared the biochar-supported K-doped g-C_3_N_4_ composites, which displayed an enhanced optical absorption in the visible region and a wider photocatalytic application scope compared to pure g-C_3_N_4_. Therefore, it is of great significance to prepare BC/g-C_3_N_4_ composites by using BC with g-C_3_N_4_ from a wide range of sources.

In this experiment, g-C_3_N_4_ was prepared by high temperature pyrolysis with urea as the precursor, and a novel decontamination material BC/g-C_3_N_4_ was synthesized by the impregnation method, which is expected to have both adsorptive and photocatalytic capabilities, contributing to the cleanup of pollutants in soil media. The surface morphology and chemical structure of BC/g-C_3_N_4_ were characterized by means of SEM, BET, FT-IR, and XRD. Meanwhile, the influence of various external factors on the removal efficiency of petroleum hydrocarbons and the possible reaction mechanism were discussed. Aside from the intrinsic adsorptive property of BC materials, the BC/g-C_3_N_4_ composites were thought to have a light-responsive capability due to the existence of graphitic g-C_3_N_4_. We expect that the BC/g-C_3_N_4_ can be a value-added biomass-derived material for the removal of petroleum hydrocarbons from soil by synergistic adsorption and photocatalysis. The aim of this study is to demonstrate the feasibility of preparing the BC/g-C_3_N_4_ composites using the proposed method, and to evaluate the unknown performances of the resultant materials for the remediation of petroleum-contaminated soil.

## 2. Materials and Methods

### 2.1. Materials

Anhydrous sodium sulfate, urea, petroleum ether, calcium chloride, isopropanol, ascorbic acid, and ethylene diamine tetraacetic acid were all analytical reagents. Coconut shell carbon was purchased from Henan Gongyi Wanjiajing Environmental Protection Material Co., Ltd. (Henan, China). It was obtained by pyrolyzing coconut shells in a quartz tube furnace at 600 °C for 6 h at a heating rate of 10 °C/min in a nitrogen (N_2_) atmosphere, and passing through a 120-mesh sieve. The soil samples were taken from the petrochemical production area of Shandong Haihua Group Co., Ltd. (Shandong, China). After collection, the soil samples were dried naturally for a week and then sieved through a 20-mesh sieve to remove large particles. The total petroleum hydrocarbon (TPH) content of the soil samples was 16,000 mg/kg, and the pH was about 7.

### 2.2. Preparation of Materials

Preparation of g-C_3_N_4_: 10g of urea was placed in a ceramic crucible with a cover, which was heated in a muffle furnace from 20 °C to 550 °C at a rate of 5 °C/min and held constant for 4 h. The samples were ground after the crucible cooling to room temperature, and g-C_3_N_4_ was obtained after passing through a 150-mesh standard sieve.

Preparation of BC/g-C_3_N_4_: the BC and g-C_3_N_4_ were mixed in proportion, and the deionized water was added into the mixed samples. Afterward, the mixed samples were ultrasonically cleaned for 30 min and stirred for 2 h at 40 °C to form the suspension. Then, the suspension was dried at 105 °C to a constant weight, and the BC/g-C_3_N_4_ composites were obtained. The doping ratios of BC to g-C_3_N_4_ were 2:1, 1:1, 1:2, 1:3, 1:4, and 1:5, and the different proportions of BC/g-C_3_N_4_ samples were obtained using the above methods.

### 2.3. Characterization Analysis of BC and BC/g-C_3_N_4_

The surface morphologies of BC/g-C_3_N_4_ and BC were obtained by a Zeiss Ultra 55 Scanning Electron Microscope (SEM, Zeiss, Sigma, Oberkochen, Germany). The specific surface areas of the samples were taken by N_2_ adsorption-desorption using the Brunauer–Emmett–Teller method (BET, Micromeritics, asap246, Norcross, GA, USA). Fourier transform infrared spectroscopy (FT-IR) spectra was obtained by a Nicolet 5700 Series infrared spectrometer (Thermofisher, Nicolet 5700, Waltham, MA, USA). Samples were characterized by X-ray diffraction spectroscopy (XRD, BrukerAXS, D8-02, Karlsruhe, Germany) for phase identification.

### 2.4. Adsorption and Photodegradation Experiments

The adsorption experiments were conducted in the dark. Petroleum-contaminated soil samples (16 g) and 1.28 g BC or g-C_3_N_4_ were mixed in a petri dish with a diameter of 9 cm. The thickness of the mixed samples was about 1.4 mm, and the moisture content of the mixed samples were maintained at 10% by regularly adding the deionized water. The adsorption experiments were maintained at 25 °C, and the soil samples without the absorbent were used as a blank control. The soil samples were taken periodically, and then passed through a 60-mesh sieve to remove BC or g-C_3_N_4_, and the TPH content of the soil was detected by an ultraviolet spectrophotometer (UV-5100B, Shanghai Leewen Scientific Instrument Co., Ltd., Shanghai, China) with petroleum ether as the extraction agent.

The photodegradation experiments were carried out with the g-C_3_N_4_ as the photocatalyst under the irradiation of a LED light. The light was placed at the top of the incubator, and the intensity of the light was 8000 Lux. One blank sample was treated with light avoidance. The procedure of the photodegradation experiments was the same as the adsorption experiments, except for the LED light irradiation.

The adsorption and photodegradation experiments of BC/g-C_3_N_4_ were carried out under light irradiation, and the experiment procedure was the same as the photodegradation experiments. The soil samples without BC/g-C_3_N_4_ were used as a blank control group. To investigate the optimal conditions of BC/g-C_3_N_4_ treating the contaminated soil, the doping ratio, dosage, soil moisture content, and soil acidity–alkalinity were chosen as the influencing factors.

Setting of influencing factors: (1) Doping ratio: 2:1, 1:1, 1:2, 1:3, 1:4, 1:5; (2) The dosage of BC/g-C_3_N_4_: 0.01, 0.02, 0.04, 0.08, 0.12, 0.16 g/g; (3) Soil acidity and alkalinity: acidic, neutral, alkaline; (4) Soil moisture content: 2%, 5%, 10%, 20%, 30%; (5) Reaction time: 3, 5, 7, 9, 11, 13, 15, 17, 19, 21, 23, 25, 27, 29 d.

### 2.5. Free Radical Trapping Experiments

To investigate the mechanism of the photodegradation, inhibition tests on pure BC/g-C_3_N_4_ samples were carried out using different radical scavengers. Isopropanol (IPA), ascorbic acid (AA), and ethylene diamine tetraacetic acid (EDTA) were chosen as the scavengers for superoxide radical (·O_2_^−^), hydroxyl radical (·OH), and hole (h^+^), respectively. The concentration of the scavengers was 1 mM.

### 2.6. Analysis of n-Alkanes and PAHs in Soil

Changes of petroleum components in soil samples were analyzed after allowing adsorption and photodegradation for 28 d. Samples were compared with those treated by BC or g-C_3_N_4_ alone.

In order to analyze the changes of nC_10_-nC_40_ in soil, the soil sample was mixed with the diatomite to dewater, and transferred into the extraction tank for extraction with the extraction repeated three times. All of the extraction liquid was collected into a concentration cup and concentrated to 1.0 mL. Then, the concentrated solution was passed through a magnesium silicate column, and the column was eluted with n-hexane of 12 mL. The concentrate and eluent were collected together and concentrated to 1 mL. The sample was then analyzed with a gas chromatography method (GC, Agilent Technologies 7890B, Palo Alto, CA, USA).

In order to analyze the changes of TPH in soil, the soil sample was mixed with anhydrous sodium sulfate and extracted by carbon tetrachloride for 0.5 h in the ultrasound instrument and 1 h in the oscillator. The extraction liquid was condensed to 1 mL in a rotary evaporator and separated by the silica gel column chromatography using dichloromethane as an eluent. The eluent was evaporated to dryness under a nitrogen atmosphere. The sample was then analyzed with a gas chromatography–mass spectrometry method (GC–MS, Agilent 7890B-5977B, Palo Alto, CA, USA).

### 2.7. Statistical Analysis

Each treatment was carried out in triplicate. The data are represented by the averages of three values, and the statistical analysis was performed using SPSS 22.0. One-way ANOVA (*p* < 0.05) was used for statistical analysis of the data obtained from the experiment, and the results of the experiments were presented with the standard deviation (SD) shown by an error bar. The letters are used to indicate whether there is a significant difference between the two data sets.

## 3. Results and Discussion

### 3.1. Characterization Analysis

#### 3.1.1. SEM Analysis

The surface morphology and microstructure of the prepared samples were investigated by SEM. As shown in Figure 1a, the microscopic morphology of BC is a tightly-arranged, banded, porous structure. The SEM image of BC/g-C_3_N_4_ is shown in Figure 1b, and the g-C_3_N_4_ was evenly distributed on the BC surface. In addition, the roughness and porosity of BC/g-C_3_N_4_ were increased, which was due to the g-C_3_N_4_ being well stacked on the BC matrix. This was beneficial for increasing the adsorption and photocatalytic efficiency of the material.

#### 3.1.2. BET Analysis

The N_2_ adsorption–desorption isotherms and pore size distribution for the BC and BC/g-C_3_N_4_ are shown in Figure 2. The N_2_ BET surface area of the BC/g-C_3_N_4_ was as high as 612.69 m^2^/g, which was higher than that of BC (541.61 m^2^/g). After loading g-C_3_N_4_ on the surface of BC, the specific surface area of BC approximately doubled, indicating that the BC/g-C_3_N_4_ had more pore structures, which was also consistent with the result of SEM. The larger specific surface area was beneficial for the enrichment of pollutants, and provided sufficient catalytic sites for photocatalytic degradation [35]. However, the average pore size did not vary much. The average pore sizes of BC/g-C_3_N_4_ and BC were 2.03 nm and 3.30 nm, respectively. The adsorption isotherms of the BC/g-C_3_N_4_ showed type I isotherms (IUPCA classification), which was mainly monolayer adsorption. The adsorption isotherms of the BC showed type II isotherms, which was mainly multilayer absorption [36]. The loading of g-C_3_N_4_ on the surface of BC increased the specific surface area and pore structure, which facilitated the adsorption and photocatalytic degradation of TPH.

#### 3.1.3. FTIR and XRD Analysis

Figure 3a represents the FTIR spectra of BC, g-C_3_N_4_, and BC/g-C_3_N_4_. For the BC sample, the band at 1520 cm^−1^ was due to the bending vibration of amino. The band at 1140 cm^−1^ was attributed to the stretching vibration of C-O in cellulose, hemicellulosic and lignin or C-O-C in cellulose and hemicellulosic, and the one at 665 cm^−1^ was attributed to the bending vibration peak of C-H of the aromatic ring. For the g-C_3_N_4_ and BC/g-C_3_N_4_ samples, an adsorption peak at 3200 cm^−1^ was observed, which originated from the stretching vibrations of N-H and O-H. In addition, the g-C_3_N_4_ and BC/g-C_3_N_4_ samples presented a series of bands of the typical stretching vibration modes of C-N heterocycles (1600, 1480, and 1260 cm^−1^) [37] and the intense bending vibration mode of the tri-s-triazine unit (800 cm^−1^) [38]. The peak shape of BC/g-C_3_N_4_ was basically the same as those of g-C_3_N_4_, and the characteristic peaks of BC/g-C_3_N_4_ were weakened. This phenomenon indicated that the BC-loaded g-C_3_N_4_ was not a simple physical mixing but formed a compact structure with lower energy [28].

Figure 3b shows the XRD diffraction pattern of BC, g-C_3_N_4_, and BC/g-C_3_N_4_. The diffractograms of g-C_3_N_4_ and BC/g-C_3_N_4_ showed two peaks at 2θ = 13.5° and 27°, corresponding to two lattice planes ((100) and (002)) of g-C_3_N_4_, respectively, indicating that g-C_3_N_4_ was successfully loaded on the surface of BC.

### 3.2. Analysis of Adsorption and Photocatalytic Capacity

The removal efficiency of TPH in the contaminated soil samples by BC, g-C_3_N_4_, and BC/g-C_3_N_4_ is shown in Figure 4. The results showed that the BC removed the TPH from the soil mainly by adsorption, and 25.7% of TPH was removed in the 28 d experiment. The TPH removal rate of g-C_3_N_4_ was only 5% under dark conditions, whereas the TPH removal rate of g-C_3_N_4_ reached 27.9% under light conditions, indicating that g-C_3_N_4_ degraded TPH in soil mainly by photocatalysis. It is clearly evident that BC/g-C_3_N_4_ showed a higher removal potential than the BC and g-C_3_N_4_ (*p* < 0.05), as 54.5% of TPH was removed in the 28-d experiment, and the removal efficiency of TPH by BC/g-C_3_N_4_ was 2.12 times and 1.95 times of the BC and g-C_3_N_4_, respectively. This is because the BC/g-C_3_N_4_ had a larger specific surface area and a wider energy band, which facilitated the adsorption of TPH on its surface and the separation of electrons and holes, thus greatly improving the TPH removal efficiency of BC/g-C_3_N_4_ [39,40].

### 3.3. Removal of TPH in Soil by BC/g-C_3_N_4_

#### 3.3.1. Effect of the BC to g-C_3_N_4_ Doping Ratio on TPH Removal Efficiency

Figure 5a shows the changes in TPH removal efficiency with different doping ratios of BC to g-C_3_N_4_. The results show that the TPH removal rate increased from 35% to 54% when the doping ratio changed from 2:1 to 1:3. However, as the g-C_3_N_4_ dosage continued to increase, the TPH removal efficiency gradually decreased. This may be due to the initially large amount of g-C_3_N_4_ adsorbed on the surface of BC, which increased the surface area of BC and provided more adsorption sites for TPH. At the same time, g-C_3_N_4_ played a role in the photocatalytic degradation of TPH in soil, and the removal rate of TPH by BC/g-C_3_N_4_ was greatly improved. When the g-C_3_N_4_ continued to increase, excessive g-C_3_N_4_ accumulated on the BC/g-C_3_N_4_ surface and occupied adsorption sites on the BC/g-C_3_N_4_ surface, showing competition with the adsorption of TPH. In addition, excessive g-C_3_N_4_ agglomerated on the surface of BC, reducing the contact area between the pollutant and the catalyst [41]. This was not conducive to the removal of TPH in the soil by BC/g-C_3_N_4_. 

#### 3.3.2. Effect of Soil Acidity and Alkalinity on TPH Removal Efficiency

Figure 5b shows the effect of soil acidity and alkalinity on the TPH removal efficiency by BC/g-C_3_N_4_. Both acidic soil and alkaline soil were not conducive to the TPH removal efficiency by BC/g-C_3_N_4_. A neutral environment was the most conducive to the removal of TPH in soil by BC/g-C_3_N_4_. The higher the concentration for H^+^ in soil under acidic conditions, the more H^+^ ions competed in the active sites of BC/g-C_3_N_4_ with TPH, which resulted in the reduction of TPH removal efficiency [42]. Under alkaline conditions, the organic acids and acidic substances of the petroleum reacted with the OH^−^ ions in soil to produce surface-active substances, making TPH easily desorbed from BC/g-C_3_N_4_, which was not favorable to the adsorption and enrichment of TPH by BC/g-C_3_N_4_ [43]. Therefore, under neutral soil conditions, BC/g-C_3_N_4_ was more beneficial to the removal of TPH in soil.

#### 3.3.3. Effect of Moisture Content on TPH Removal Efficiency

Figure 5c shows the effect of moisture content of soil on the TPH removal efficiency. It can be seen from the figure that the TPH removal efficiency gradually increased with the increasing of moisture content. Low moisture content was not conducive to the functioning of BC/g-C_3_N_4_, and the TPH removal rate of air-dried soil was only 40%. On the one hand, the moisture content of soil was so low that the electrons excited by visible light could only be transferred by air as the medium, and the transfer efficiency was not as good as in water. Thus, the TPH removal efficiency was low. The mass transfer rate of the solid–solid phase was much lower than that of the solid–liquid phase, so the low moisture content also inhibited the removal of TPH by BC/g-C_3_N_4_. When the moisture content of soil reached 10%, the TPH removal rate reached a maximum value of about 58% and then remained constant.

#### 3.3.4. Effect of BC/g-C_3_N_4_ Dosage on TPH Removal Efficiency

Figure 5d shows the effect of the dosage of BC/g-C_3_N_4_ on the TPH removal efficiency. It can be seen from the figure that the dosage of BC/g-C_3_N_4_ increased from 0.01 g/g to 0.02 g/g, and the TPH removal rate increased rapidly from 44.5% to 50.8%. More adsorption sites were provided for TPH in the soil with the increase in the dosage of BC/g-C_3_N_4_, and more oxidation active intermediates were produced for the photocatalysis of TPH in the reaction system. However, with the excessive increase of BC/g-C_3_N_4_, the adsorption sites and oxidation active substances were surplus, which made the specific removal of TPH decrease. As shown in Figure 5d, when the dosage of BC/g-C_3_N_4_ increased from 0.01 g/g to 0.16 g/g, the specific removal amount of TPH decreased rapidly from 354.1 mg/g to 32.1 mg/g. Therefore, the optimal dosage of 0.08 g/g can not only achieve a better removal effect, but also control the reasonable production cost.

### 3.4. Kinetic Analysis

The experimental data were fitted by pseudo-first-order kinetic and pseudo-second-order kinetic [44]. The fitting results are shown in Figure 6 and Table 1.
(1)$$\text{The pseudo-first-order kinetic model: }q_{t}=q_{e}\left({1-e}^{{-K}_{1}t}\right)$$
(2)$$\text{The pseudo-second-order kinetic model: }q_{t}=\frac{q_{e}^{2}K_{2}t}{{1+q}_{e}K_{2}t}$$
where *q_e_* is the equilibrium removal capacity of TPH, mg/g; *t* is the reaction time, d; *K*_1_ is the rate constant of the pseudo-first-order kinetic model, d^−1^; *K*_2_ is the rate constant of the pseudo-second-order kinetic model, mg·g^−1^·d^−1^.

Figure 6 and Table 1 show the pseudo-first-order kinetic model, pseudo-second-order kinetic model, and fitting parameters of the TPH removal by BC/g-C_3_N_4_, respectively. As shown in Figure 6, the removal rate of TPH from soil by BC/g-C_3_N_4_ increased gradually with time at first. After a period of time, the removal rate of TPH gradually reached the equilibrium state. It can be seen from Table 1 that the *R*^2^ of the pseudo-second-order kinetic model was larger than that of the pseudo-first-order kinetic model, and the theoretical removal capacity calculated by the pseudo-second-order kinetic model was closer to the actual value, so the pseudo-second-order kinetic model was more suitable for the process of TPH removal. Therefore, the removal of TPH in soil by BC/g-C_3_N_4_ was not only a simple physical adsorption, but also included a chemical process [45].

### 3.5. Gas Chromatography Analysis

The contents of n-alkanes and PAHs in soil were determined by GC and GC-MS methods. According to the test results, the total concentration of PAHs in the original soil sample was 0.63 mg/kg, and only benzo(g,h,i)perylene, benzo(k)fluoranthene, dibenz(a,h)anthracene, and indeno(1,2,3-cd)pyrene were detected. There were no PAHs that could be detected after the soil samples were treated by BC, g-C_3_N_4_ or BC/g-C_3_N_4_. Figure 7a shows the GC chromatogram of n-alkanes in soil samples treated by different materials. As observed from the chromatogram of the original soil sample, the n-alkanes were composed of a wide range of hydrocarbons from nC_13_ to nC_35_. The relative concentrations of nC_17_-nC_28_ were higher than the other n-alkanes, and that of C_28_ was the highest. The peaks of nC_13_ and nC_14_ almost disappeared with the contaminated soil treated by BC, g-C_3_N_4_, or BC/g-C_3_N_4_. In addition, the peaks of nC_29_-nC_35_ also almost disappeared with the contaminated soil treated by BC/g-C_3_N_4_. As shown in Figure 7b, the removal rates of nC_13_-nC_15_ were above 95% with the contaminated soil treated by BC or g-C_3_N_4_. The removal rates of nC_16_-nC_35_ were around 50% for BC treatment, which reached around 80% for g-C_3_N_4_ treatment. When BC/g-C_3_N_4_ was used to remove n-alkanes from soils, the nC_13_-nC_15_ were completely removed, and the removal rates of the nC_16_-nC_35_ increased to more than 90%. The BC/g-C_3_N_4_ synergistically worked to improve the removal of petroleum hydrocarbons from the soil significantly, which was consistent with the previous results shown in Figure 4.

In addition to PAHs and n-alkanes, the TPH in contaminated soil also included branched alkanes and cycloalkanes. The removal of n-alkanes by g-C_3_N_4_ photocatalysis was higher than that by BC adsorption, but the difference in the removal of TPH was not significant, indicating that the removal of branched alkanes and cycloalkanes by BC adsorption was more effective. This is because the structure of cycloalkanes was stable and it was difficult to open their rings, so the g-C_3_N_4_ photocatalytic degradation was difficult to perform. However, the adsorption of BC was not selective and was little-influenced by the nature of the pollutants. Thus, the removal rates of different components in TPH by BC were more average. The removal rates of n-alkanes treated by g-C_3_N_4_ and BC/g-C_3_N_4_ were much higher than the TPH removal rates in soil samples. This indicated that g-C_3_N_4_ and BC/g-C_3_N_4_ were relatively ineffective in removing branched and cyclic alkanes from the TPH, and that the remaining constituents in the soil were mostly obstinate-branched and cyclic alkanes [46].

### 3.6. Photocatalysis Mechanism

Figure 8 shows the adsorption and photocatalytic mechanism of BC/g-C_3_N_4_. The g-C_3_N_4_ could generate electron–hole pairs under visible light. Under irradiation, electrons were excited from the valence band (VB) of g-C_3_N_4_ to its conduction band (CB), leaving a positively-charged hole (h^+^) in the VB of g-C_3_N_4_. The disadvantage of the pure g-C_3_N_4_ photocatalytic material was that the photo-generated electrons and holes rapidly combined to fail, which affected its photocatalytic efficiency [47]. As a conductive channel, BC enhanced the electron transfer rate of g-C_3_N_4_ and provided more active sites for photocatalysis [48]. In addition, BC played the role of electron traps, and the photo-generated electrons of g-C_3_N_4_ could be transferred to BC for temporary storage, which was conducive to the separation of electron–hole pairs, so that more electrons were involved in the photocatalytic process and the photocatalytic efficiency was improved [49]. The rich pore structure and large specific surface area of BC could enrich TPH in soil, and had a synergistic effect with the photocatalytic process of g-C_3_N_4_.

The electrons in g-C_3_N_4_ and the electrons accumulated on BC combined with O_2_ to form superoxide radical (·O_2_^−^), whereas the remaining holes in the VB combined with H_2_O to form hydroxyl radical (·OH). The removal of petroleum hydrocarbons from the soil is carried out by the strong oxidizing properties of these groups. In order to understand the degradation mechanism in depth, the active radicals were identified by free radical capture experiments. Isopropanol (IPA), ascorbic acid (AA), and ethylene diamine tetraacetic acid (EDTA) were employed as the hydroxyl radical (·OH), superoxide radical (·O_2_^−^), and hole (h^+^) scavengers. As Figure 9 shows, the removal rates of TPH decreased when the scavengers were added, indicating that ·OH, ·O_2_^−^, and h^+^ were involved in the photocatalytic degradation of TPH. In particular, the removal rate of TPH greatly decreased from 55.94 to 24.87% after the addition of AA, indicating that ·O_2_^−^ was the main active radical in the photocatalytic process.

## 4. Conclusions

An environmentally-friendly soil remediation method based on the BC and g-C_3_N_4_ has been developed to remediate petroleum-contaminated soil. The novel composites BC/g-C_3_N_4_ were prepared by the impregnation method. It can be confirmed that the BC/g-C_3_N_4_ was successfully formed from the surface composition and morphological structure. The results of the adsorption and photodegradation experiments showed that the composite had superior adsorption and photocatalytic properties. Under the visible light irradiation, the BC/g-C_3_N_4_ showed excellent performance for removing the TPH from soil. The removal rate of TPH by BC/g-C_3_N_4_ was 2.12 and 1.95 times that of BC and g-C_3_N_4_. As a conductive channel, BC enhanced the electron transfer rate of g-C_3_N_4_ and provided more active sites for photocatalysis. In addition, the porous structure of BC can provide more adsorption sites for TPH. These effects synergistically improved the adsorption and photocatalytic activity of BC/g-C_3_N_4_ significantly. The optimum doping ratio of 1:3 between BC and g-C_3_N_4_ was determined by single-factor experiments, and the best removal effect could be achieved by adding BC/g-C_3_N_4_ at a dosing rate of 0.08 g/g in a neutral soil with 10% moisture content. The active species trapping experiments reveal that the ·O_2_^−^ plays a major role in the process of TPH degradation. The BC/g-C_3_N_4_ composites could be considered a promising material for the degradation of organic pollutants. This study provides a potential idea for the efficient degradation of TPH in soil. However, this method also faces the problem of difficult catalyst recovery in practical applications. In the future, compounding with magnetic nanomaterials should be considered, to use magnetism to recover composite materials from soil, so as to achieve the reuse of materials and to reduce costs. In addition, it is important to explore the co-doping or tri-doping of composites with certain elements to further improve their light absorption range and electron transfer capability.

## Figures and Tables

**Figure 1 ijerph-19-08290-f001:**
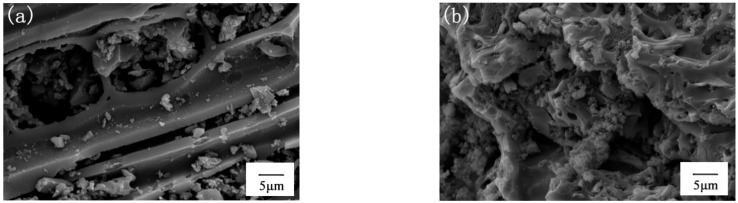
SEM images of (**a**) BC and (**b**) BC/g-C_3_N_4_ (1:3).

**Figure 2 ijerph-19-08290-f002:**
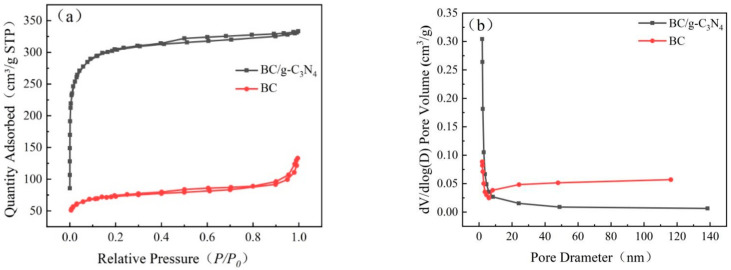
(**a**) N_2_ adsorption–desorption isotherms; (**b**) pore size distribution curves of BC and BC/g-C_3_N_4_ (1:3).

**Figure 3 ijerph-19-08290-f003:**
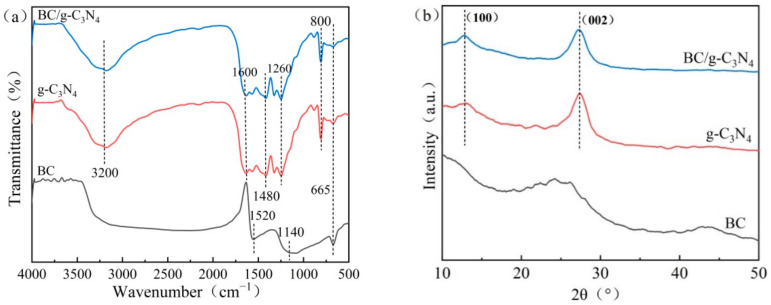
(**a**) FTIR and (**b**) XRD of BC, g-C_3_N_4_, and BC/g-C_3_N_4_ (1:3).

**Figure 4 ijerph-19-08290-f004:**
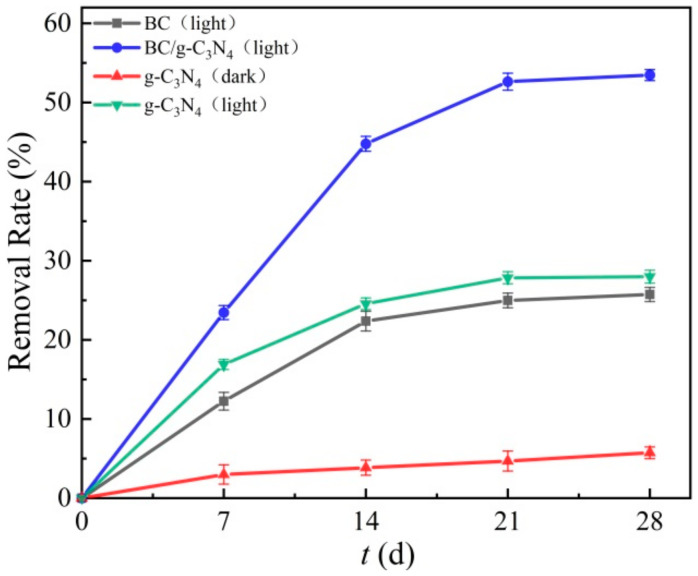
TPH removal efficiency by BC, BC/g-C_3_N_4_ (1:3), g-C_3_N_4_ (dark), g-C_3_N_4_ (light).

**Figure 5 ijerph-19-08290-f005:**
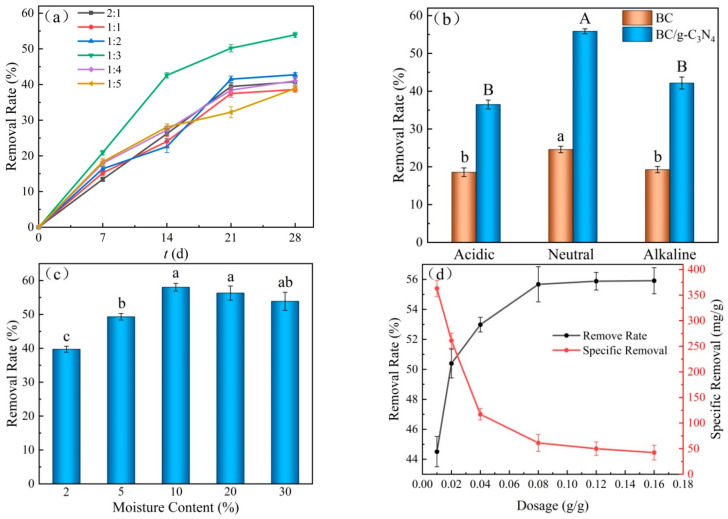
The TPH removal efficiency by BC/g-C_3_N_4_ under various conditions: (**a**) doping ratio, (**b**) pH, (**c**) moisture content, (**d**) dosage.

**Figure 6 ijerph-19-08290-f006:**
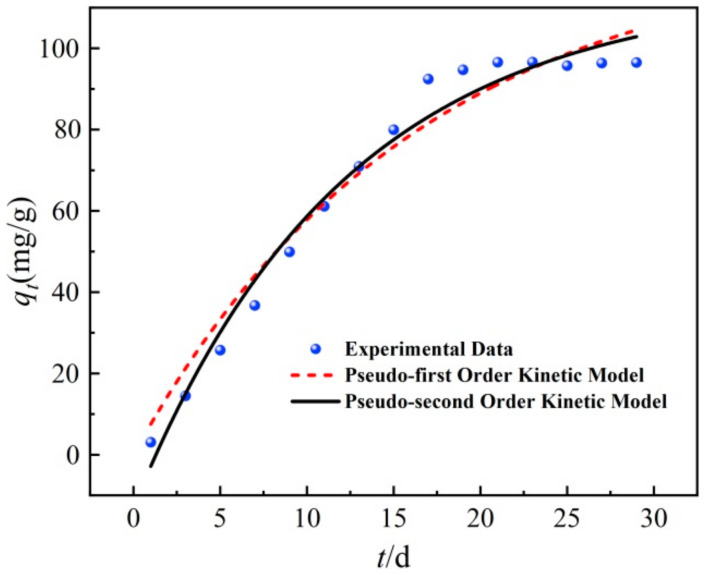
Kinetics model of TPH removal by BC/g-C_3_N_4_.

**Figure 7 ijerph-19-08290-f007:**
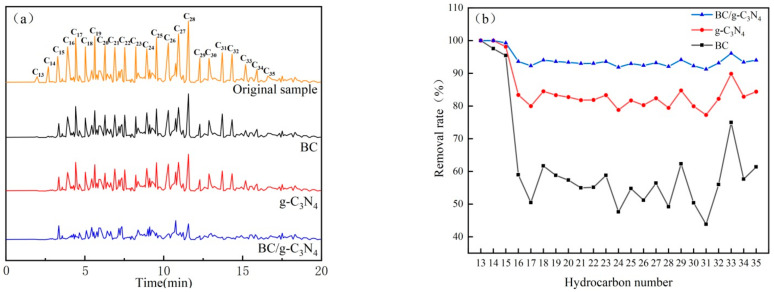
(**a**) GC chromatogram of n-alkanes in soil samples treated with different materials at 28 d; (**b**) The removal rate of each n-alkane in soil samples treated with different materials.

**Figure 8 ijerph-19-08290-f008:**
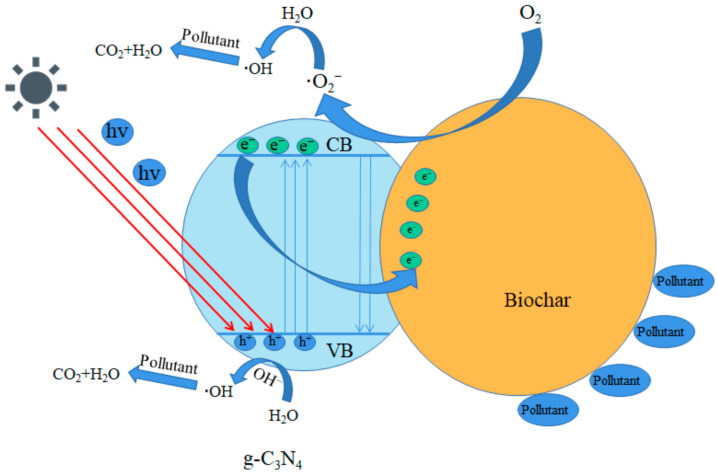
Removal mechanism of TPH by BC/g-C_3_N_4_ (1:3).

**Figure 9 ijerph-19-08290-f009:**
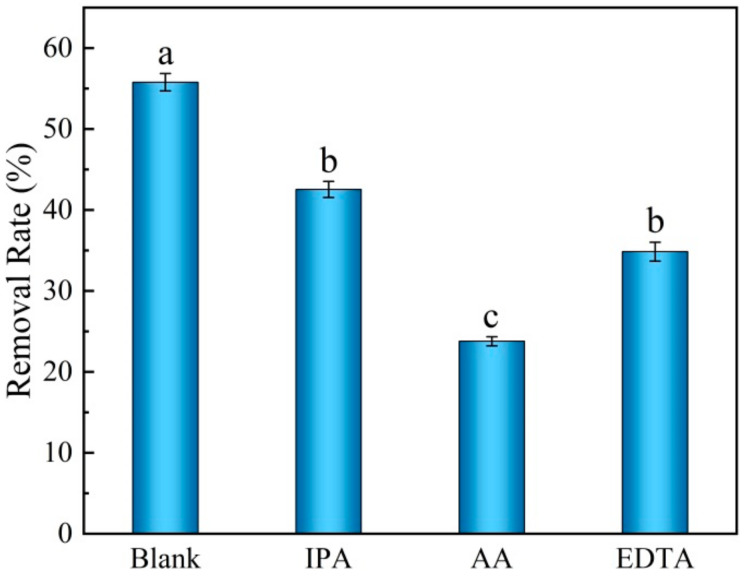
BC/g-C_3_N_4_ (1:3) toward TPH degradation in the presence of different trapping agents.

**Table 1 ijerph-19-08290-t001:** Kinetic parameters of TPH removal by BC/g-C_3_N_4_.

Pseudo-First-Order Kinetics Model			Pseudo-Second-Order Kinetics Model		
*q_e_*/(mg·g^−1^)	*K*_1_/d^−1^	*R* ^2^	*q_e_*/(mg·g^−1^)	*K*_2_/(mg·g^−1^·d^−1^)	*R* ^2^
102.51	0.0620	0.959	101.72	0.0161	0.981

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
