# Peer review of "Study on the Enhanced Remediation of Petroleum-Contaminated Soil by Biochar/g-C3N4 Composites"

_ijerph, 2022, doi:10.3390/ijerph19148290_

Round 1

Reviewer 2 Report

Manuscript Title: Study on the Enhanced Remediation of Petroleum-contaminated Soil by Biochar/g-C3N4 Composites

Manuscript ID: ijerph-1781030

The manuscript written by Lin et al. reports quite interesting results. In this work, enhanced remediation of petroleum-contaminated soil by biochar/g-C3N4 composites was investigated. The critical discussion of adsorption and photocatalytic mechanism of BC/g-C3N4 and removal mechanism are noteworthy features of this manuscript. Due to the widespread use of petroleum and its toxic effects on human beings, there is a need to conduct the studies related to removal process of these compounds. The data presented in this manuscript will be of great interest to the readers and extend to a high degree of our knowledge about petroleum remediation.

In totality, the conceptualization, designing of experiments and the overall write up is good and quite clear. However, it needs some corrections and there are some minor queries which the authors should kindly respond to make it good, to the best of my knowledge.

Technical queries/ suggestions:

1. There are many abbreviations in the manuscript.I suggest the authors can add a section after the manuscript text to describe the abbreviations. In addition, ensure that abbreviations/acronyms are defined the first time they appear in each of three sections: the abstract; the main text; the first figure or table.

2. Abstract can be written more precisely and explain novelty of this work.

3. Introduction section should briefly place the study in a broad context and highlight why it is important. It should define the purpose of the work and its significance. The current state of the research field should be reviewed, and key publications cited. Highlight controversial and diverging hypotheses when necessary. Finally, briefly mention the main aim of the work and highlight the principal conclusions. However, the novelty and significance of the manuscript were not highlighted. 

4. Statistical analysis is very important. I suggest the authors add a new section in the Materials and Methods to describe the details of the statistical analysis.

5. Figure 4. TPH removal efficiency by BC, BC/g-C3N4 (1:3), g-C3N4(dark), g-C3N4(light). Does this experiment repeat trials? Please add analysis of variance and statistical analysis.

6. Figure 5. The TPH removal efficiency by BC/g-C3N4 under various conditions: (a) doping ratio, (b) pH, (c) moisture content, (d) dosage. Need statistical analysis.

7. Figure 9. BC/g-C3N4 (1:3) toward TPH degradation in the presence of different trapping agents. Add the statistical analysis.

8. Authors should revise the Conclusion section for the better understanding of the topic and its future research.

9. Line 392: Please use italic for the genus names Pseudomonas aeruginosa. Check throughout the manuscript.  

10. Line 40: adsorption, incineration [8], chemical oxidation, leaching, phytoremediation and microbial-remediation method. Add references to support this statement for each method respectively.

Reviewer 3 Report

The submitted manuscript represents an important issue in terms of  “Study on the Enhanced Remediation of Petroleum-contaminated Soil by Biochar/g-C3N4 Composites”. Comments are provided below.

Keywords: the word biochar should be entered

Line 43: when using the abbreviation for the first time, it should be explained

Line 43: biochar can be produced from many different materials, not only biomass. For this purpose, for example, various types of waste materials (i.e. 10.1016/j.jaap.2015.07.011) or sewage sludge (i.e. doi.org/10.1016/j.jclepro.2020.122259) or mixtures of different materials are used (i.e. 10.1016/j.jcou.2018.10.019). Please complete this

Line 46-53: where are the modifications to improve the adsorption of petroleum compounds?

Line 75-81: The aim of the research and the research hypothesis are missing

Line 92: no methodology for biochar production. Raw material, temperature, carrier gas type, pyrolysis duration? All of this is of great importance for the properties of biochar

L113: Was soil moisture kept at the same level throughout the experiment or was it only initially determined and then not controlled?

Line 247: there is no information in the methodology that different levels of moisture were used

Round 2

Reviewer 2 Report

The authors have considered all comments and revised the manuscript accordingly based on these comments. The revision is fine and can be accepted for publication in current form.

Reviewer 3 Report

Accept